# Inhibition of Solar UV-Induced Matrix Metalloproteinase (MMP)-1 Expression by Non-Enzymatic Softening Cherry Blossom (*Prunus yedoensis*) Extract

**DOI:** 10.3390/plants10051016

**Published:** 2021-05-19

**Authors:** Yeong-A Jung, Ji-Yoon Lee, Pomjoo Lee, Han-Seung Shin, Jong-Eun Kim

**Affiliations:** 1Department of Food Science and Biotechnology, Dongguk University-Seoul, Ilsandong-gu, Goyang-si 10326, Korea; yzoo13@naver.com; 2Biomodulation Major, Department of Agricultural Biotechnology, Seoul National University, Seoul 08826, Korea; jylee.dec@gmail.com; 3RAFIQ Cosmetics Co., Ltd., Cheonan-si 31094, Korea; kuempire@gmail.com; 4Department of Food Science and Technology, Korea National University of Transportation, Chungju-si 27909, Korea

**Keywords:** cherry blossom, non-enzymatic softening technology, anti-photoaging, functional material

## Abstract

Cherry blossom (*Prunus yedoensis*) petals are used as ingredients in many cosmetics. However, despite their use in numerous products, the exact function of cherry blossom petals in cosmetics is unclear. Therefore, we need evidence-based studies to support the labeling claims that are made in cherry blossom products in the cosmetics industry. We investigated the skin anti-aging potential of non-enzymatic softening cherry blossom extract (NES-CBE) in this study. The extract desalinated, to improve its quality such that it can be used as a functional material for the skin. The anti-wrinkle effect of NES-CBE was investigated on human keratinocytes (HaCaT cells) under solar UV (sUV) light exposure. We found that NES-CBE reduced the sUV-induced matrix metalloproteinase (MMP)-1 expression and modulated the transactivation of the activator protein (AP)-1. Furthermore, NES-CBE suppressed the phosphorylation of MEK1/2 and ERK proteins, indicating its regulation of sUV-induced MAPK signaling. Additionally, we observed NES-CBE reduced MMP-1 protein expression in a human skin equivalent model. Taken together, these results suggest that NES-CBE reduces sUV-induced MMP-1 protein expression through reducing AP-1 transactivation via regulation of the MEK1/2-ERK pathway.

## 1. Introduction

Solar ultraviolet (sUV) light irradiation plays a major factor in aging, with an estimated 80% of facial skin photoaging caused by sUV light [1,2]. The primary symptoms of photoaging include wrinkles, a leathery texture, rough pigmentation, laxity and dryness [3]. sUV is divided into a wavelength spectrum that comprises three subtypes including UVA (320–400 nm), UVB (280–320 nm), and UVC (200–280 nm). The penetration of UVC is effectively prevented by the Earth’s ozone layer, while both UVA and UVB can reach and penetrate the facial skin [4]. sUV irradiation consisting of about 94.5% UVA and 5.5% UVB usually causes skin wrinkles and chronic exposure causes skin damage and aging [5].

Metalloproteinase (MMP) is an important enzyme that breaks down the extracellular matrix. Currently, about 20 types are known [3]. Among them are three enzymes that degrade collagen: MMP-1, 8, and 13. MMP-1 is known as a rate limiting enzyme in the decomposition of collagen caused by sUV in the skin. MMP-1 plays critical roles in degrading types I and III collagens, which are structural components of the ECM [4]. MMP-1 is overexpressed after UV exposure, and reports suggest that skin wrinkles are alleviated when its expression is blocked [4]. Therefore, the inhibition of MMP-1 protein expression to prevent wrinkles can be used as a potential therapeutic strategy to reduce photoaging [6].

Activator protein (AP)-1 is a key transcription factor of sUV induced MMP-1 [6]. AP-1 is a heterodimer which plays crucial roles in inflammation related disease [7]. The AP-1 transcription factor includes JUN, FOS/FRA, ATF/CREB, MAF and the JDP subfamilies [6]. MAPK and Akt pathways mediate sUV-induced AP-1 activity [6]. Therefore, the regulation of these pathways is an excellent strategy to inhibit UV-induced MMP-1 expression [3].

Cherry blossom *(Prunus yedoensis*) petals have been used as a popular ingredient in cosmetics [8]. Previous studies show that *P. yedoensis* flower extract exerts various physiological effects, including anti-cancer, anti-inflammatory, and antioxidant effects [9]. Furthermore, the petals contain a lot of physiologically active compounds, such as phenolic and terpenoid compounds [10]. However, due to their short harvest time and shelf life, cherry blossoms are usually preserved with salt, making formulation into cosmetics difficult. Thus, cherry blossoms have been added in only very small amounts or as synthetic flavors in cosmetics. Thus, the beneficial effects of cherry blossom on skin health remain unclear, although many products with cherry blossom are commercially available.

The purpose of this study was to demonstrate the anti-wrinkle effect of non-enzymatic softening cherry blossom (*P. yedoensis*) extract (NES-CBE, Figure 1). We elucidated the inhibitory effects of NES-CBE on sUV-induced MMP-1 expression in human keratinocytes (HaCaT). This was used as a human skin equivalent model, as an alternative to testing in animals. We also investigated the molecular mechanisms behind transactivation of AP-1 and inhibition of sUV-induced MMP-1 protein expression. The purpose of this study was to demonstrate the anti-wrinkle effect of NES-CBE. We elucidated the inhibitory effects of NES-CBE on sUV-induced MMP-1 expression in human keratinocytes (HaCaT). These were used as a human skin equivalent model—an alternative to testing in animals. We also investigated the molecular mechanisms underlying the transactivation of AP-1 and the inhibition of sUV-induced MMP-1 expression.

## 2. Results

### 2.1. Antioxidant Effects and Contents of Total Phenolics in NES-CBE

Antioxidant activity can improve skin health. It prevents aging of the skin by blocking the various cell signaling pathways provoked by sUV radiation. ABTS and DPPH assays were performed to determine antioxidant properties. The antioxidant activity of NES-CBE was compared with that of vitamin C (Table 1). We show that NES-CBE exhibited vitamin C equivalents of NES-CBE in ABTS and DPPH assays. A major antioxidant is phenolics [11]. We found the total phenolic content in NES-CBE (Table 1). NES-CBE exhibited tannic acid equivalents.

### 2.2. Effects of NES-CBE on sUV-Induced MMP-1 Expression

sUV is the most important cause of skin aging [6]. Too much substrate degradation by sUV-induced MMP-1 contributes the connective tissue damage which occurs during skin aging [3]. MMP-1 usually initiates fibrillar collagen cleavage, leading to skin wrinkles [12]. To show NES-CBE inhibition of MMP-1 in HaCaT cells, the inhibitory effects on sUV-induced MMP-1 protein expression were measured (Figure 2A,B). The MTT assay results showed that NES-CBE did not exhibit cytotoxicity at concentrations up to 100 μg/mL (Figure 2C). The cells were pre-treated with indicated concentrations of NES-CBE for 1 h and then irradiated with sUV to the cells at 25 kJ/m^2^. To determine the MMP-1 protein in the culture medium after sUV irradiation, the medium was tested by western blotting. Our results showed that MMP-1 protein expression levels had increased upon sUV irradiation, and NES-CBE treatment reduced sUV-induced MMP-1 protein levels. We used MMP-2 as a loading control (Figure 2A,B).

### 2.3. Effects of NES-CBE on sUV-Induced MMP-1 Promoter Activity by Reducing AP-1 Transactivation

We investigated MMP-1 transcriptional activity and AP-1 transactivation. Luciferase reporter gene assays showed that sUV increased MMP-1 promoter activity levels and AP-1 transactivation levels, and NES-CBE treatment lowered these transcriptional and transactivation activities (Figure 3A,B). AP-1 is an important transcription factor in sUV-induced MMP-1 expression. sUV drives the transcriptional activity of AP-1, which increases the expression of MMP-1 [12]. Our results suggest that NES-CBE inhibits sUV-induced MMP-1 expression by suppressing AP-1 activation.

### 2.4. Effects of NES-CBE on sUV-Induced ERK1/2 Pathways

The ERK mitogen-activated protein kinase (MAPK) pathway plays a pivotal role in reducing AP-1 transactivation [13,14]. In our studies, the phosphorylation levels of ERK1/2 and MEK1/2 were increased by sUV irradiation, and NES-CBE treatment reduced the sUV-induced MEK1/2 and ERK1/2 phosphorylation (Figure 4A). However, NES-CBE did not modulate the Akt, p38, and JNK1/2 signaling pathways (Figure 4B). These results indicate that NES-CBE suppresses sUV-induced MMP-1 protein expression by preventing AP-1 transactivation by directly inhibiting the ERK1/2 pathway.

### 2.5. Effects of NES-CBE on sUV-Induced Skin Damage and MMP-1 Protein Expression in Human Skin Equivalent

UV radiation damages the epidermis and leads to rapid photoaging [3]. To show the physiological relevance of NES-CBE mediated inhibition of MMP-1, we examined the effect of NES-CBE on sUV-induced MMP-1 protein expression in a human skin equivalent model (Figure 5). The cell extracts were collected 2 days after sUV irradiation. Then, the paraffin-embedded human skin equivalent was sectioned. As expected, the epidermal layer was degraded in the sUV-treated human skin equivalent tissue and NES-CBE recovered the damaged skin tissue. Similar to the HaCaT cells (Figure 3), the MMP-1 protein levels in sUV-treated human skin equivalent tissue were increased. NES-CBE significantly inhibited sUV-induced MMP-1 protein expression. 40 μg/mL of NES-CBE reduced the sUV-induced MMP-1 protein expression to a level that was comparable to the control group. These results suggest that in a preclinical setting, NES-CBE may have anti-wrinkle effects in skin, by suppressing sUV-induced MMP-1 protein expression

## 3. Discussion

Repeated exposure to sUV radiation accelerates premature aging of the skin or photoaging [15]. Photoaging in skin is a part of the senescence process. It is associated with sagging, laxity, and wrinkles [3]. Skin wrinkles are mainly caused by sUV that increase MMP-1 protein expression and oxidative stress and deplete collagen in the skin [6]. sUV-induced MMP-1 regulation directly impacts the skin wrinkling process [12]. Overexpression of MMP-1 by sUV irradiation facilitates skin wrinkling by disrupting tissue integrity [16]. Previous studies have shown that UV-activated MAP kinase pathways, including JNK and p38, lead to AP-1 activation, which subsequently enhances MMP-1 expression. sUV activates MAPK and Akt pathways [6]. Notably, UV radiation can activate the ERK signaling pathway, which plays an important role in the activation of AP-1 transcription factors [16]. The AP-1 activity is enhanced by the MAPK and Akt pathways through the phosphorylation of c-Fos and c-Jun, because AP-1 consists of c-Jun homodimer or c-Jun and c-Fos heterodimer [6]. In epidermal and dermal cells, AP-1 forms complexes to regulate the transcription of MMP-1, which is an important transcription factor. The upregulation of MMP-1 mRNA is modulated by AP-1 [6]. The photoaging characterized by skin wrinkles is a naturally occurring process of senescence. Thus, it is important to find a novel agent that can inhibit or delay the process of photoaging [16].

Cherry blossom extract has been studied to have many beneficial effects for health, such as antioxidant, anti-inflammatory, and anti-cancer activities [9,10,17]. Extracts from *P. yedoensis* have been shown to confer antioxidant effects [9]. The bark, leaves, and fruits have been reported to cause inflammation [17]. However, previous studies have been not conducted to improve skin health using cherry blossom extract because the salt accumulation and browning of cherry blossom petals makes formulation difficult. Therefore, evidence-based studies are needed to support the labeling claims made in cherry blossom products in the cosmetics industry. In this study, we hypothesized that cherry blossom petals (desalinated and minimized for loss of functional substances) are beneficial for skin health. Thus, we tried to improve the quality of the desalinated cherry blossom (*P. yedoensis*) petals extract by using a non-enzymatic softening process [18,19]. To our knowledge, that the beneficial effects of desalinated and softened cherry blossom petals on skin health have not been reported. Therefore, for the first time, we evaluated the effects of NES-CBE on photoaging as a skin health indicator. We aimed to evaluate the anti-wrinkle effect of NES-CBE for the development of functional materials for skin health by investigating its ability to suppress sUV-induced MMP-1 expression.

In most studies, UVA and UVB have been studied independently. However, for better physiological relevance, it is better to use UV of the same wavelength as sunlight [20]. We used a UVA-340 lamp that emitted UV radiation consisting of 94.5% UVA and 5.5% UVB (15). Then, we irradiated 25 kJ/m^2^ of sUV for HaCaT cells and a human skin equivalent model, which corresponds to an average UV daylight dose in New York City [21]. sUV irradiation markedly increased MMP-1 expression in HaCaT cells.

In summary, these results suggest that NES-CBE inhibits sUV-induced MMP-1 expression by suppressing the transactivation of AP-1 through the inhibition of the MEK1/2-ERK pathway. Therefore, we propose that NES-CBE is a potential novel anti-wrinkle agent that may prevent photoaging when used in formulations of cosmetic products.

## 4. Materials and Methods

### 4.1. Chemicals

Salted cherry blossom petals were purchased from Chinriu Honten Limited (Kanagawa, Japan). Fetal bovine serum (FBS), Trypsin-EDTA, Penicillin–streptomycin solution, and Dulbecco’s Modified Eagle’s medium (DMEM) were purchased from Welgene (Gyeongsan, Korea). Antibodies against phosphorylated ERK 1/2 at the Thr202/Tyr204 residues and total ERK1/2 were obtained from Santa Cruz Biotechnology (Santa Cruz, CA, USA). The MMP-1 antibody was purchased from R&D Systems Inc. (Minneapolis, MN, USA). All other antibodies were obtained from Cell Signaling Technology (Beverly, MA, USA). The lentiviral expression vector, pGF-AP1-mCMV-EF1-Puro was purchased form System Biosciences (Palo Alto, CA, USA). pGF-MMP-1-mCMV-EF1-puro vector was developed in our previous studies [22]. psPAX and pMD2.0G were obtained from Addgene Inc. (Cambridge, MA, USA). A chemiluminescence detection kit was purchased from Dainbio (Sungnam, Korea).

### 4.2. Preparing NES-CBE

Salted cherry blossom petals were washed, and distilled water (20×) was added. After soaking at 25 °C for 30 min, the distilled water was replaced, and the procedure was repeated four times. Then, these desalinated cherry blossom petals were mixed with a cellulose softening agent (2×). Soluble components were extracted at 80 °C for 90 min. The extract was filtered using Whatman filter paper number 2 (Whatman, Maidstone, UK) and then freeze-dried (Figure 1).

### 4.3. ABTS Radical Scavenging Activity

The method was described in our previous study [11]. Briefly, 1.0 mM AAPH was mixed with 2.5 mM ABTS as diammonium salt in water. The mixture was incubated in the dark for 30 min, and the optical absorbance was measured using a microplate reader (BioTek. Winooski, VT, USA) at 734 nm.

### 4.4. DPPH Radical Scavenging Activity 

The was method described in our previous study [11]. The DPPH radical was dissolved in ethanol. The sample solutions were added to the DPPH radical solution. The mixture was incubated in the dark for 30 min. Subsequently, the optical absorbance was measured using a microplate reader (BioTek) at 517 nm.

### 4.5. Total Phenolic Content

The method was described in our previous study [11]. Briefly, Folin-Ciocalteu phenol reagent was added. After 10 min, Na_2_CO_3_ solution (7%) was added and mixed. After incubating in room temperature for 90 min, the absorbance was measured using a microplate reader (BioTek) at 750 nm. The total phenolic content of the samples was expressed in grams per milligram of tannic acid equivalents (TAE).

### 4.6. Cell Culture

HaCaT cells were obtained from CLS Cell Lines Services GmbH (Heiderberg, Germany) and cultured in DMEM with 1% (*v/v*) penicillin/streptomycin and 10% (*v/v*) FBS at 37 °C under 5% CO_2_.

### 4.7. sUV Irradiation

We used the UVA-340 lamps as our sUV source (Q-Lab Corporation, Cleveland, OH, USA). The UVA-340 lamps produce sunlight-like UV light ranging from 295 nm to 365 nm, with an emission peak at 340 nm. The percentages of UVA and UVB were 94.5% and 5.5%, respectively. HaCaT cells and the human skin equivalents were irradiated with sUV (25 kJ/m^2^).

### 4.8. Cell Viability

Cell viability of NES-CBE was measured using MTT assays. HaCaT cells were cultured in 96-well plates, diluted in serum-free media, and treated with NES-CBE. After 48 h, the cells were added to the MTT solution (0.5 mg/mL) and incubated at 37 °C for 3 h. DMSO (200 μL) was then added to each well to dissolve the formazan. The absorbance was measured using microplate reader (570 nm, BioTek).

### 4.9. Western Blot

When HaCaT cells reached 70% confluent the cells were starved in serum-free DMEM overnight. Then they were treated with 10, 20, and 40 μg/mL of NES-CBE for 1 h, followed by irradiation with sUV. The medium was harvested and centrifuged at 16,000× *g* for 15 min. Cell lysates were prepared using cell lysis buffer (50 mM Tris-HCl at pH 8.0, 1% NP-40, 0.1% SDS, 0.5% deoxycholate, 1 mM dithiothreitol, 1 mM phenylmethylsulfonyl fluoride, 1 mM sodium vanadate, and 0.15 M NaCl). Bio-Rad protein Assay Kit (Bio-Rad, Feldkirchen, Germany) was used for protein concentrations. The proteins were separated by electrophoresis using a 10% SDS–polyacrylamide gel. The proteins were sub-transferred onto polyvinylidene fluoride membranes (PVDF, Merck Millipore, Burlington, MA, USA). The membranes were blocked with 5% bovine serum albumin or non-fat milk for 2 h. These were then incubated with specific primary antibodies for 4 h. HRP-conjugated secondary antibodies (Cell Signaling Technology, Beverl, MA, USA) were added to washed membranes. An ECL detection kit (Dainbio, Korea) was used for visualizing protein bands. The western blot data were analyzed using densitometric analysis in Image Studio software (LI-COR, Lincoln, NE, USA).

### 4.10. Gelatin Zymography

To measure MMP-2, gelatin zymography was performed by including 0.1% gelatin (*w/v*) in 10% SDS polyacrylamide gels. The protein samples were mixed with a loading buffer for zymography (0.25 M Tris buffer at pH 6.8, containing 25% glycerol, 10% SDS, and 0.1% bromophenol blue). The gel was then washed 3 times with renaturing buffer (Koma Biotech,Seoul, Korea) at room temperature for 60 min, and incubated in developing buffer (Koma Biotech) at 37 °C for 48 h. The gel was visualized with 0.5% Coomassie Brilliant Blue.

### 4.11. Luciferase Reporter Gene Assay

pGF-MMP-1-mCMV-EF1-puro and pGF-AP-1-mCMV-EF1-Puro vectors with the packaging vectors (psPAX and pMD2.0G) were transfected into 293T cells using jetPEI transfection reagent as per the manufacturer’s instructions. The medium was changed after 12 h transfection. After 24 h, the medium was prepared. A 0.45 μm syringe filter was used for preparing viral particles. It injected 60% of the confluent HaCaT into the HaCaT cells overnight with 8 μg/mL of polybrene (Santa Cruz Biotechnology). The cell culture medium was replaced with fresh growth medium. The cells could recover for 24 h before they were subjected to puromycin (InvivoGen, Pak Shek Kok, Hong Kong, 2 mg/mL) selection for 36 h. The HaCaT cells that were selected after transduction were cultured for 48 h and starved for 24 h in serum-free media. Subsequently, the cells were treated with indicated concentrations of NES-CBE for 1 h, followed by irradiation with sUV. After either 24 h (MMP-1) or 12 h (AP-1), cell extracts were prepared using lysis buffer for luciferase (Promega, Madison, WI, USA). Transactivation was determined using a luciferase reporter gene assay kit (Promega) complying with the manufacturer’s instructions.

### 4.12. Human Skin Equivalent Preparation

The human skin equivalent (Neoderm-ED, TEGO Science, Seoul, Korea) was treated with NES-CBE (10 and 40 μg/mL) for 1 h and irradiated with sUV twice daily for four days. The medium was changed every 2 d during incubation at 37 °C under 5% CO_2_. The media were prepared for MMP-1 validation. Masson’s trichrome staining was used as described in a previous study [23].

### 4.13. Statistical Analysis

The data were statistically analyzed using R software (R Foundation). The differences between the control and the sUV-irradiated control were assessed using the Student’s *t*-test. One-way ANOVA followed by Duncan’s multiple range test was used to compare the differences between the sUV-exposed groups. A probability value (*p*) of less than 0.05 was considered statistically significant.

## Figures and Tables

**Figure 1 plants-10-01016-f001:**
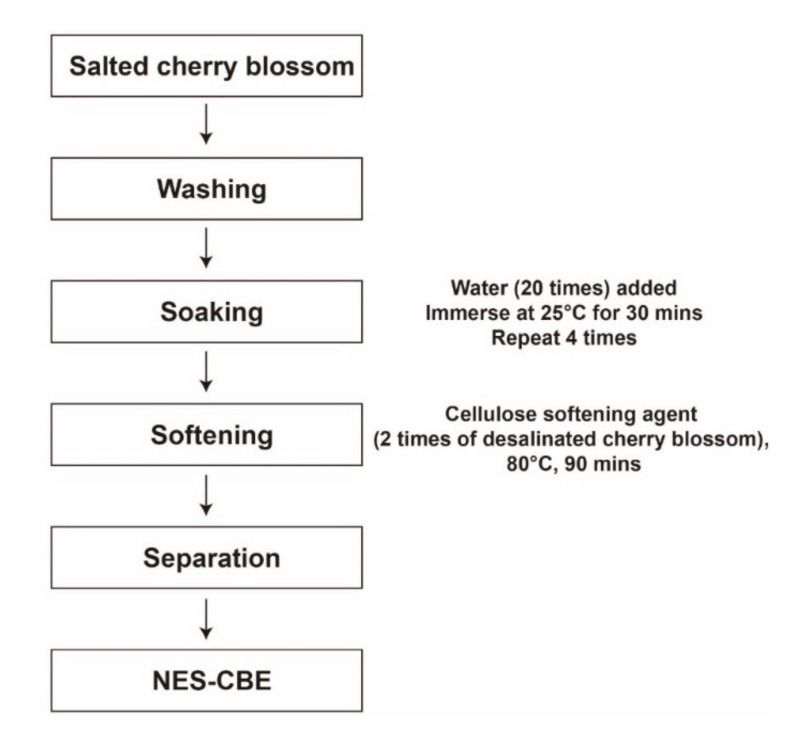
NES-CBE extraction process from salted cherry blossom using desalination and softening technologies.

**Figure 2 plants-10-01016-f002:**
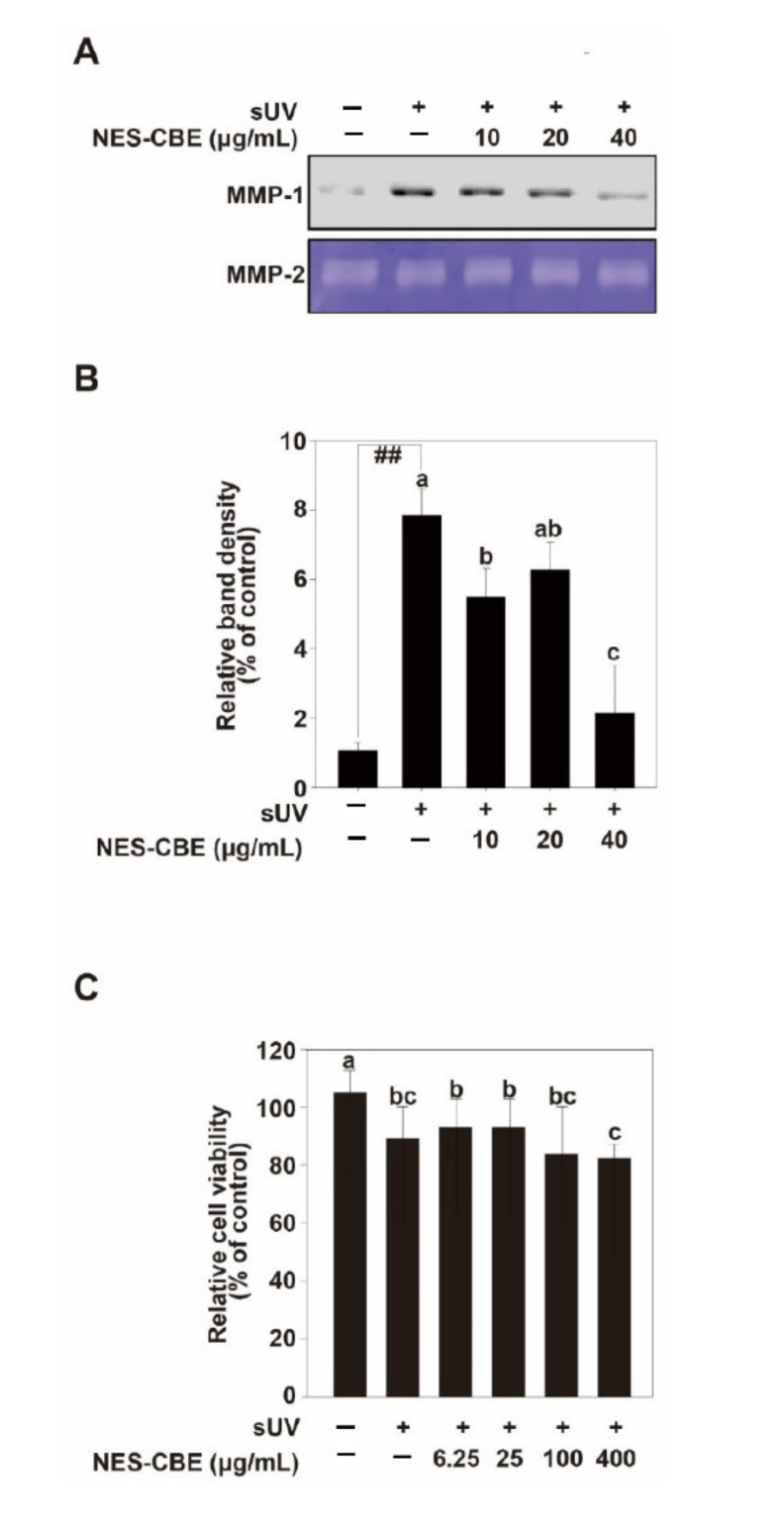
Effect of NES-CBE on solar ultraviolet (sUV)-induced matrix metalloproteinase (MMP)-1 protein expression. (**A**,**B**) HaCaT cells were seeded and incubated at 80% confluent. The cells were starved for 24 h. NES-CBE at the indicated concentrations was pretreated for 1 h, and then further irradiated with 25 kJ/m^2^ sUV. The cells were incubated for 48 h at 37 °C. The medium was prepared for western blot. Western blot assay was used for analyzing protein expression. MMP-2 was a loading control. The bands were quantified using NIH imageJ software. Data (*n* = 3) are presented as mean ± SD. The experiment was performed in triplicates and bars marked with different letters (**a**–**c**) are significantly different (*p* < 0.05, ^##^
*p* < 0.01), relative to the control cells. (**C**) Effect of NES-CBE on HaCaT cell viability. A percentage of cell viability above 80% is considered as non-cytotoxic; within 80–60% as weakly cytotoxic; within 60–40% as moderately cytotoxic and below 40% as strongly cytotoxic. The MTT assay results showed that NES-CBE did not exhibit cytotoxicity until a concentration of 100 µg/mL. Data (*n* = 5) shown are presented as means ± SD. The experiment was performed in triplicate and bars marked with different letters (**a**–**c**) are significantly different, relative to the control cells, one-way ANOVA. Duncan’s Multiple Range test is used as a post hoc test to measure specific differences.

**Figure 3 plants-10-01016-f003:**
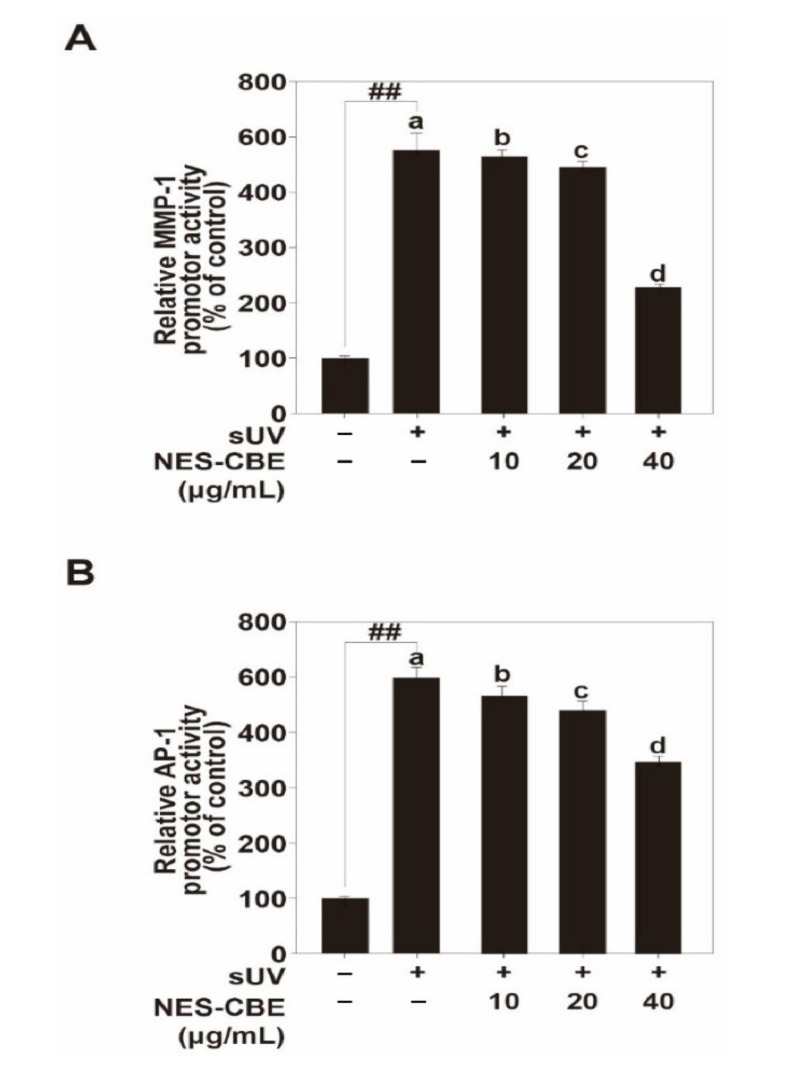
NES-CBE suppresses sUV-induced MMP-1 transcription by inhibiting AP-1 transactivation. (**A**) Effect of NES-CBE on sUV-induced MMP-1 promotor activity. MMP-1 promoter reporter plasmid transduced HaCaT cells were prepared, and MMP-1 promotor activity was measured, described in materials and methods. (**B**) Effect of NES-CBE on AP-1 promoter transactivation activity induced by sUV. HaCaT transduced with an AP-1 reporter plasmid transduced HaCaT cells was used, and AP-1 transactivation was measured described in materials and methods using a luciferase reporter gene assay. Data (n = 5) represent the mean values ± SD. Means with letters (a–d) in a graph are significantly different from each other at *p* < 0.05 (^##^
*p* < 0.01), relative to the control cells. Different letters indicate a significant difference at *p* < 0.05, one-way ANOVA. Duncan’s Multiple Range test was used as a post hoc test to measure specific differences.

**Figure 4 plants-10-01016-f004:**
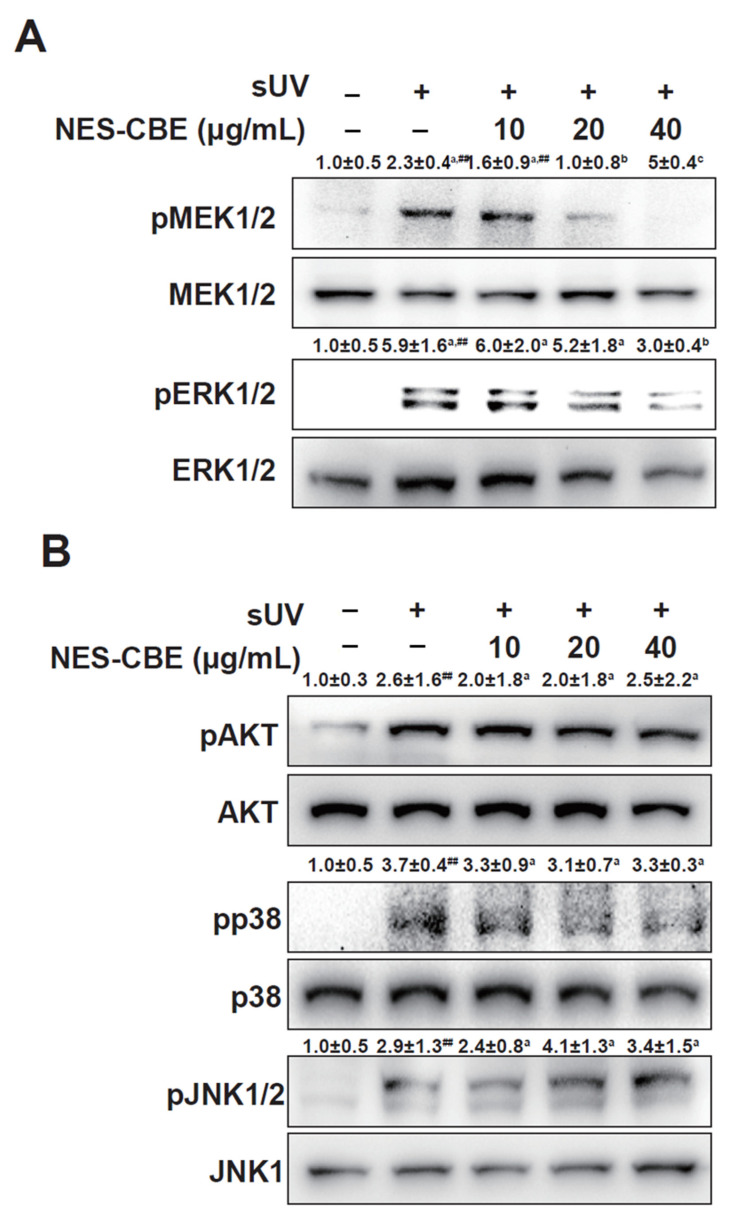
Inhibitory effect of NES-CBE on the sUV-induced MEK/ERK signaling pathway. (**A**) Effect of NES-CBE on the sUV-induced ERK1/2 signaling pathway. (**B**) Effect of NES-CBE on the sUV-induced Akt, p38 and JNK1/2 signaling pathways. HaCaT cells were treated with NES-CBE at the indicated concentration for 1 h, treated with sUV for 30 min. The cell lysates were prepared for western blotting. Image Studio software (LI-COR, Lincoln, NE, U.S.A.) was used for band density measurement. Bands are normalized to that of non-phospho form. Data are means ± SD of at least three independent experiments. ^##^ Significant difference between the control and sUV-treated groups (*p* < 0.01), one-way ANOVA. Duncan’s Multiple Range test was used as a post hoc test to measure specific differences. Different letters indicate a significant difference at *p* < 0.05.

**Figure 5 plants-10-01016-f005:**
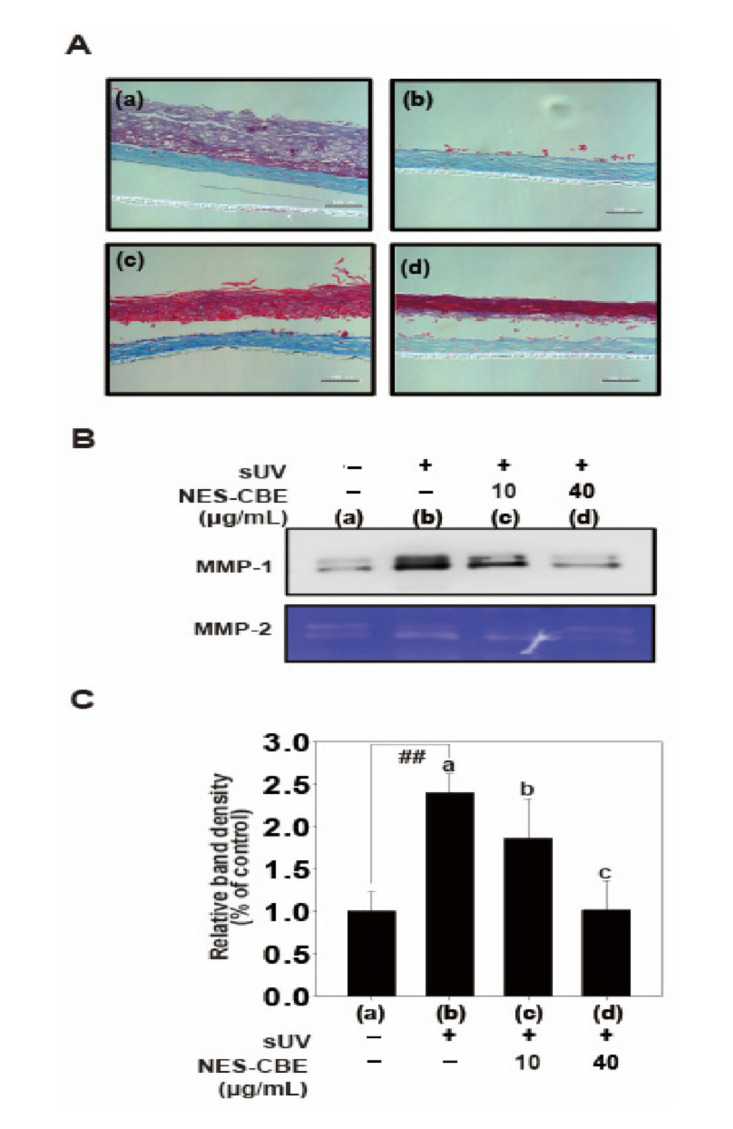
(**A**) NES-CBE inhibits sUV-induced skin damage and MMP-1 protein expression in a human skin equivalent model. (**B**) Human skin equivalent tissue was pretreated with different concentrations (10 to 40 μg/mL) of NES-CBE for 1 h and then exposed to 25 kJ/cm^2^ sUV twice a day for two days at 2 h intervals. 24 h after the last sUV irradiation, the medium was harvested for MMP-1 expression. The human skin equivalent tissue was fixed in 4% formaldehyde solution for Masson’s trichrome staining. Scale bar shown at 100 μm. (**C**) Untreated controls: (a) sUV (25 kJ/m^2^) only, (b) sUV and NES-CBE (10 μg/mL), (c) sUV and (d) NES-CBE (40 μg/mL). Image Studio software (LI-COR) was used for band density measurement. The experiment was performed in triplicates and bars marked with different letters (a–c) are significantly different (*p* < 0.05, ## *p* < 0.01), relative to the control cells with one-way ANOVA. Duncan’s Multiple Range test was used as a post hoc test to measure specific differences.

**Table 1 plants-10-01016-t001:** Antioxidant effect of NES-CBE.

Method	ABTS	DPPH	Total Phenolic Content
Comparison	Vitamin C equivalents	Vitamin C equivalents	tannic acid equivalents
Unit	mg VCEAC ^(1)^/100 g	mg VCEAC/100 g	mg TAE ^(2)^/g
Value	1499.31 ± 7.64	564.64 ± 6.40	36.86 ± 0.61

^(1)^ Vitamin C Equivalent Antioxidant Capacity. ^(2)^ Tannic acid equivalents.

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
