# Peer review of "Inhibition of Solar UV-Induced Matrix Metalloproteinase (MMP)-1 Expression by Non-Enzymatic Softening Cherry Blossom (Prunus yedoensis) Extract"

_plants, 2021, doi:10.3390/plants10051016_

Round 1

Reviewer 1 Report

The paper entitled “Inhibition of solar UV-induced matrix metalloproteinase (MMP)-1 expression by non-enzymatic softening cherry blossom (Prunus yedoensis) extract” is an attempt to evaluate the real biological value of cherry blossom extracts often used in cosmeceutical products.

The work is consistent, the experiments satisfactorily carried on and explained, but some revisions are necessary before it can be published.

The general use of English language is good but a general check of typos is necessary (as examples see page 9, lines 163 and 165);

page 2, lines 75-89 is a duplicated paragraph;

The “Discussion” section needs a deep revision; actually it represents more a further introduction to the topic than a discussion concerning the specific results of the authors in the described experiments; it could be a good idea to join the results and discussion into the same paragraph.

References should be checked and improved; some points need references more focused, such as ref. 1 and 2 that appears more as forced and redundant self-citations.  

Author Response

Reviewer 1:

Comment 1: The paper entitled “Inhibition of solar UV-induced matrix metalloproteinase (MMP)-1 expression by non-enzymatic softening cherry blossom (Prunus yedoensis) extract” is an attempt to evaluate the real biological value of cherry blossom extracts often used in cosmeceutical products. The work is consistent, the experiments satisfactorily carried on and explained, but some revisions are necessary before it can be published. The general use of English language is good but a general check of typos is necessary (as examples see page 9, lines 163 and 165); page 2, lines 75-89 is a duplicated paragraph;

Response 1: We revised our manuscript as your suggestion. We corrected the abbreviation.

Comment 2: The “Discussion” section needs a deep revision; actually it represents more a further introduction to the topic than a discussion concerning the specific results of the authors in the described experiments; it could be a good idea to join the results and discussion into the same paragraph. References should be checked and improved; some points need references more focused, such as ref. 1 and 2 that appears more as forced and redundant self-citations.

Response 2: We revised our discussion part and change references.

Comment 2: Figure 3: Please note that the scale bars are too small to be visible in print. - This point has not been addressed and the scale bars are still too small.

Response 2:We revised our manuscript as your suggestion. We redrawed scale bar.

Comment 3:Figures 4,5: Please set ‘mL’ instead of ‘ml’. Please correct throughout the whole manuscript.- This point has not been addressed and it still says ‘pg/ml’ instead of ‘pg/mL’.

Response 3:We revised our manuscript as your suggestion.

Reviewer 2 Report

Dear Editor,

Please find bellow my comments and suggestions regarding the manuscript entitled: “ Inhibition of solar UV-induced matrix metalloproteinase (MMP)-1 expression by non-enzymatic softening cherry blossom (Prunus yedoensis) extract” by  Yeong-A Jung, Ji Yoon Lee, Pomjoo Lee, Han-Seung Shin, and Jong-Eun Kim.

From the formal point of view, the manuscript is well written and structured and the English language is quite correct. As a minor comment, I miss a photo illustrating the flower of Prunus yodoensis.

I am aware that the authors center their research on the anti-wrinkle effect of Prunus yedoensis on keratinocytes under solar UV light exposure, among other biochemical studies, in order to justify its use in dermatologic preparations for skin protection. However, there are preliminary aspects that the authors should clarify in their manuscript.

Even though they mention that, there are previous studies on the composition and natural products present in the cherry blossom, see references 9, 10 and 11, among other. It is not clear from the text, neither from the methodology section what kind of active principles are extracted by the procedure outlined in Figure 1. The composition of the extract should be analyzed by HPLC and/or HPLC-MS and other techniques. It looks like these natural products present should be water soluble and no other solvents are used (oils or organic solvents). Take into account that the human skin has a high lipidic content. On the other hand, the biological (biochemical) effects of the extract are carefully evaluated but not the biophysical ones. It is not evaluated the possibility that any component of the extract, due to its structure, having chromophores, may act as a UV filter by itself, independently of their biochemical of physiological implications.

In addition, in the methodology describing the extraction procedure it is not clear how the petals are collected, are they fresh or dried? How rapidly are processed after collection. During the previous washing with distilled water, are any natural components lost? Is the washing water analyzed? What cellulose-softening agent is used? The extraction at 80ºC for 90 min (in the presence of the oxygen of air and light) may alter the structure of the natural products present in the petals. Is this possibility evaluated? The answers to these questions are significant to assure the repeatability of the presented studies.

The biochemical methodologies used are quite correct and the studies are adequate and novel. For this reason, my opinion on the manuscript is quite positive and I recommend the publication of the manuscript after clarification of the aforementioned issues.

Author Response

Reviewer 2:

Comment 1: I am aware that the authors center their research on the anti-wrinkle effect of Prunus yedoensis on keratinocytes under solar UV light exposure, among other biochemical studies, in order to justify its use in dermatologic preparations for skin protection. However, there are preliminary aspects that the authors should clarify in their manuscript. Even though they mention that, there are previous studies on the composition and natural products present in the cherry blossom, see references 9, 10 and 11, among other. It is not clear from the text, neither from the methodology section what kind of active principles are extracted by the procedure outlined in Figure 1. The composition of the extract should be analyzed by HPLC and/or HPLC-MS and other techniques. In addition, in the methodology describing the extraction procedure it is not clear how the petals are collected, are they fresh or dried? How rapidly are processed after collection. During the previous washing with distilled water, are any natural components lost? Is the washing water analyzed? The extraction at 80ºC for 90 min (in the presence of the oxygen of air and light) may alter the structure of the natural products present in the petals. Is this possibility evaluated? The answers to these questions are significant to assure the repeatability of the presented studies.

Response 1: Salted cherry blossom petals were purchased Chinriu Honten Limited (Kanagawa, Japan).  We added this information to our manuscript. Using this, it was extracted like figure 1. The extract was prepared considering that it can be well used in cosmetics. In this study, the effect of the extract on improving skin health was studied. Through the results of this study, since the extract has sufficient efficacy, the study on what substances are contained in the extract will proceed in our next study.

Comment 2: It looks like these natural products present should be water soluble, and no other solvents are used (oils or organic solvents). Take into account that the human skin has a high lipidic content. On the other hand, the biological (biochemical) effects of the extract are carefully evaluated but not the biophysical ones. It is not evaluated the possibility that any component of the extract, due to its structure, having chromophores, may act as a UV filter by itself, independently of their biochemical of physiological implications.

Response 2: As suggested by the reviewer, the fat-soluble substance will be well absorbed. However, it is made of cosmetic materials, and water-soluble substances have advantages depending on the formulation. Clinical trials should be conducted to see if this substance actually works in humans. We planned a clinical trial. The original plan was to publish clinical trial results in this paper. However, the clinical trial was postponed due to the COVID-19 situation. NES-CBE inhibited the phosphorylation of ERK induced by sUV well, but did not inhibit the phosphorylation of Akt or JNK. If NES-CBE acts as a UV filter, all signals will have to be inhibited. Therefore, NES-CBE does not act as a UV filter.

Comment 3: The biochemical methodologies used are quite correct and the studies are adequate and novel. For this reason, my opinion on the manuscript is quite positive and I recommend the publication of the manuscript after clarification of the aforementioned issues.

Response 3:Thank you.
